# Impact of BRAF and MEK Inhibitors on Gingival Hyperplasia in Melanoma Patients—A Case Report

**DOI:** 10.3390/jcm14010065

**Published:** 2024-12-26

**Authors:** Tanja Veljovic, Milanko Djuric, Ivana Gusic, Nada Vuckovic, Bojana Ramic, Jelena Mirnic

**Affiliations:** 1Department of Dental Medicine, Faculty of Medicine, University of Novi Sad, 21000 Novi Sad, Serbia; milanko.djuric@mf.uns.ac.rs (M.D.); ivana.gusic@mf.uns.ac.rs (I.G.); bojana.ramic@mf.uns.ac.rs (B.R.); jelena.mirnic@mf.uns.ac.rs (J.M.); 2Dentistry Clinic of Vojvodina, Department of Dental Medicine, Faculty of Medicine, University of Novi Sad, 21000 Novi Sad, Serbia; 3Center for Pathology and Histology, Faculty of Medicine, University of Novi Sad, 21000 Novi Sad, Serbia; nada.vuckovic@mf.uns.ac.rs

**Keywords:** melanoma, gingival hyperplasia, BRAF inhibition, MEK inhibitors, case report

## Abstract

**Background**: Although BRAF inhibitors, such as vemurafenib, produce a marked response in patients with advanced melanoma with a BRAF V600 mutation, they eventually develop resistance to this treatment. To address this issue, vemurafenib is increasingly combined with the MEK inhibitor cobimetinib, leading to improved response rates and enhanced survival. However, this treatment modality is associated with numerous side effects. We present a case of gingival hyperplasia in a patient treated with vemurafenib, along with the strategy adopted for the management of this condition, and the impact of subsequent cobimetinib administration on its severity. **Methods**: The 59-year-old male patient in the focus of this report presented at the Department of Periodontology at the Medical Faculty, University of Novi Sad, in 2019, complaining of gingival overgrowth and bleeding. The patient reported persistent gum swelling during the preceding six months, which he ascribed to the use of vemurafenib, 960 mg twice daily, since 2018, when this medication was prescribed as a part of malignant melanoma treatment. Detailed clinical examination revealed significant gingival overgrowth around all present teeth, affecting the vestibular as well as the oral sides. The patient underwent thorough scaling and root planing, followed by the surgical removal of hyperplastic gingiva. After gingivectomy, the patient was scheduled for follow-up visits at one-month intervals. Six months after gingivectomy, vemurafenib dose was reduced to 720 mg twice daily, and cobimetinib was introduced at 60 mg per day. **Results**: The treatment protocol adopted in this study, combined with cobimetinib administration, stabilized the gingiva condition in this patient. However, due to his overall poor oral hygiene, gingiva remained inflamed and edematous, but was no longer hyperplastic and hyperkeratotic in appearance. **Conclusions**: This case underscores the importance of recognizing and adequately addressing this complication, as its adverse effect on a patient’s quality of life can potentially compromise treatment protocol adherence.

## 1. Introduction

Drug-induced gingival hyperplasia was first described in the scientific literature in 1939 [1]. After this case, which pertained to the use of the anticonvulsant drug diphenyl-hydantoin, the number of drugs associated with its occurrence gradually increased. Based on the accumulated body of evidence, three large groups of drugs that can cause gingival hyperplasia have been identified: anticonvulsant drugs, calcium channel blockers, and immunosuppressants. While the frequency of gingival enlargement caused by different drugs is difficult to determine due to significant variations across published findings, available data indicate that gingival enlargement as a result of phenytoin use occurs in about 50% of cases [2], cyclosporine in about 8–85% [3], and calcium channel blockers in about 20–83% of patients [4]. Clinical signs of gingival enlargement can be seen as early as 1–3 months after the therapy initiation [5].

Once the condition advances, it not only causes aesthetic problems but also starts to interfere with speech, mastication, and eating, and increases the risk of developing periodontitis. In most cases, gingival enlargement causes sensitivity and bleeding, pathological tooth movement, occlusion and speech problems, and tends to become painful once the gingiva covers the tooth crown.

Recent research also suggests that some drugs used in the malignant melanoma treatment can lead to gingival hyperplasia.

Melanoma is a skin tumor resulting from the malignant transformation of melanocytes. Although it accounts for less than 2% of all skin cancers, it is the key contributor to skin malignancy-related mortality, as untreated distant metastases are associated with a median survival of only 6–9 months [6,7].

Melanomas frequently metastasize, commencing with regional lymph node involvement, and subsequently spread to distant sites, such as other parts of the skin, lungs, liver, central nervous system, and bones [8].

Patients with advanced metastatic or unresectable malignant melanoma are typically offered immunotherapy and targeted therapy. While chemotherapy was previously the primary treatment choice, as it does not improve survival, it is now considered appropriate only for patients who are not candidates for the aforementioned treatment modalities and for whom no suitable clinical trials are available. The aim of targeted therapies is to inhibit the mitogen-activated protein kinase (MAPK) signaling pathway components, especially in cases of its constitutive activation due to the BRAF V600E mutation [9], which occurs in approximately 60% of melanomas [10]. Such therapies involve the intake of specific drugs, such as vemurafenib. This novel class I RAF-selective inhibitor is indicated as monotherapy in the treatment of adults with an inoperable or metastatic form of melanoma with a positive BRAF V600 mutation, as it not only improves the survival rate but has also demonstrated a 60% antitumor response rate in these patients [11].

However, as melanoma can develop resistance to vemurafenib therapy within 6–7 months, it is often combined with the MEK inhibitor cobimetinib [12]. Clinical studies have shown that this combination therapy helps counteract the mechanisms through which melanoma cells develop resistance to vemurafenib and leads to an increase in progression-free survival from 6.2 months (with vemurafenib alone) to 9.9 months [12].

Vemurafenib use is also associated with a range of adverse effects, which occur in 92–95% of patients and typically include erythema, maculopapular rash, photosensitivity, folliculitis, cutaneous squamous cell carcinoma, and keratoacanthoma, but may also manifest as erythema nodosum and toxic epidermal necrolysis [13,14,15,16,17]. Clinical evidence also indicates that most of the dermatological adverse effects are caused by the inhibition of the MAPK pathway in keratinocytes, which can result in inflammatory responses, impaired keratinocyte migration, and increased keratinocyte apoptosis [18]. While treatment discontinuation in such cases is not warranted, dose reductions or temporary therapy cessation may be required until these conditions are resolved.

More recent publications also point to a potential association between vemurafenib use and the emergence of gingival hyperplasia, with adverse impact on the patient’s quality of life owing to difficulties with mastication [19,20]. Given the limited data on this side effect, and even greater scarcity of reports on gingival hyperplasia in patients with metastatic melanoma receiving both vemurafenib and cobimetinib, this gap in the literature has motivated the present study.

## 2. Case Presentation

The 59-year-old male patient in the focus of this report presented at the Department of Periodontology at the Medical Faculty, University of Novi Sad, in 2019 complaining of gingival overgrowth and bleeding. The patient indicated persistent gum swelling for the preceding 6 months and associated this condition with his use of Zelboraf (vemurafenib), which was prescribed in 2018 for the treatment of malignant melanoma. The patient reported that, in 2016, malignant melanoma was diagnosed on his back and was surgically removed, without subsequent radiation treatment or chemotherapy. He attended check-ups at 3-month intervals. Initially, his condition remained stable, but after 12 months, metastases developed in the axillary lymph nodes, which were surgically removed. A few months later, metastases were found in the liver and spleen as well. Since March 2018, he has been taking Zelboraf 960 mg twice daily. Several months after commencing this treatment modality, he began to notice skin rash, followed by gum swelling, diarrhea, insomnia, nervousness, impaired vision (requiring changes to his ophthalmic prescription twice within a few months), and photosensitivity.

At the beginning of 2019, Zelboraf was discontinued for a month, leading to a marked reduction in gum enlargement and bleeding. However, the symptoms reappeared when the therapy resumed, and gradually became more pronounced.

A clinical examination revealed significant gingival overgrowth around all present teeth, affecting both the vestibular and oral sides. The enlarged gums bled easily on probing and were covered with erosions and white thick plaque, which was hyperkeratotic in appearance. The patient was found to have significant supra- and subgingival tooth deposits (Figure 1). A swab was taken and the subsequent microbial culture analyses confirmed the presence of the normal oral cavity microbiota.

For the eroded surfaces, regular application of Triamcinolone 0.1% in orabase was prescribed. At the control examination ten days later, erosions on most of the affected surfaces were epithelized while white plaques remained unchanged.

Due to the presence of both periodontal disease and gingival hyperplasia with abundant supragingival and subgingival dental plaque, oral hygiene instructions were given to the patient and scaling and root planing were undertaken. Scaling and root planing were performed under local anesthesia (2% lidocaine with adrenaline 1:100,000) using periodontal curettes (Gracey Access curettes, Kohler, Austria) and ultrasonic scalers (Mini Piezon, Electro-Medical Systems, Nyon, Switzerland). The therapy was carried out on two consecutive days within 24 h, starting with the right maxillary and mandibular quadrants, without the use of antibiotics or antiseptics.

One month after the scaling and root planing completion, the patient was scheduled for an incisional biopsy for histopathologic examination. Hyperplastic gingiva samples taken from two sites—the interdental papilla from the vestibular side in the area of tooth 16 and from the palatal side of the upper front teeth—were subjected to pathohistological analysis.

The histological report revealed stratified squamous epithelium with acanthosis, focal parakeratosis, and minor erosions. Keratinocytes exhibited vacuolated cytoplasm. In the lamina propria, there was dense infiltration of chronic inflammatory cells, including numerous granulocytes, along with dilated capillary blood vessels (Figure 2). These findings were consistent with gingival hyperplasia.

By the next follow-up examination, which took place four weeks later, gingival inflammation had decreased. However, as gingiva remained hyperplastic, the decision was made to surgically remove the enlarged tissue (Figure 3). Gingivectomy was performed under local anesthesia, during which the hyperplastic gingiva was removed, starting with the vestibular and palatal side of the upper jaw, followed by the lower jaw. A COE-PAK™ surgical dressing was placed over surgical sites, and was removed at the control visit seven days later. The intervention was performed without any complications and the postoperative course was uneventful.

After gingivectomy, the patient was scheduled for follow-up visits at one-month intervals. Inadequate oral hygiene, tooth deposits, and slightly edematous inflamed gums with bleeding on probing were observed at each visit. Initially, there were no signs of gingival overgrowth, but in a couple of months, gingiva started swelling again (Figure 4).

Six months after gingivectomy Zelboraf dose was reduced to 720 mg twice daily, and Cotellic (cobimetinib) was introduced at 60 mg per day, which resulted in a decrease in gingival enlargement (Figure 5).

The treatment protocol remained unchanged until the last follow-up appointment, approximately three years after the initial visit. At this stage, gingiva condition was stable. Due to poor oral hygiene, gingiva remained inflamed and edematous, but was no longer hyperplastic and hyperkeratotic in appearance. Unfortunately, the patient soon passed away due to the underlying disease.

## 3. Discussion

Mangold et al. [19] were the first to document gingival hyperplasia in a patient undergoing vemurafenib treatment for malignant melanoma. Similar to our patient, their study subject experienced a resolution of the gingival hyperplasia during 1-month temporary vemurafenib withdrawal. The authors attributed this adverse outcome to the drug’s cutaneous side-effect profile that, according to Rinderknecht et al. [21], overlaps with the dermatological manifestations of genetic disorders associated with activating germline RAS mutations (RASopathies), such as cardiofaciocutaneous syndrome (CFC), Costello syndrome (CS), Noonan syndrome, and hereditary gingival fibromatosis [22,23]. Accordingly, they proposed that the presence of germline RASopathies could be seen as a risk factor for the side effects associated with RAS/MAPK pathway activation. Mangold et al. [19] thus concluded that gingival hyperplasia observed in their patient might represent an additional RASopathic adverse effect of vemurafenib therapy.

In another report on gingival hyperplasia, Salman et al. [20] also ascribed it to vemurafenib use. However, they proposed an alternative mechanism for this association, hypothesizing that it involves increased PI3K–mTOR pathway activation, which could occur during the course of treatment, given that secondary resistance to vemurafenib develops in nearly all patients after about 6 months of therapy. Salman et al. supported this argument by highlighting that gingival hyperplasia is associated with tuberous sclerosis complex and Cowden syndrome, in which mutations leading to mTOR pathway activation are well established [24]. This hypothesis was further substantiated by the emergence of gingival hyperplasia later in the treatment regimen, as was the case in our patient.

The mechanisms underpinning the relationship between gingival hyperplasia and drugs that give rise to this condition are complex and insufficiently explored. Therefore, it is presently not possible to identify patients at risk of developing this condition following the use of specific drugs [25]. Nonetheless, empirical evidence suggests that men are more prone to gingival hyperplasia than women. It is also more frequently noted in young people, and is more common in the lower jaw than in the upper jaw. Findings published in pertinent literature further indicate that the degree of inflammation and gingival enlargement largely depend on the type of drug used, the prescribed dose and therapy duration, but also the patient’s oral hygiene status. Individual susceptibility (including genetic factors) and environmental factors also play an important role in the emergence of this condition [26].

Drug-induced gingival enlargement has similar clinical characteristics, whereby it usually affects the gingiva on the labial side of the front teeth. The condition first occurs in the area of the interdental papilla after which the expansion occurs laterally. The first signs of hyperplasia are localized nodular enlargements of the interdental papilla, whereby existing inflammation is exacerbated in the presence of conditions that are favorable for dental plaque accumulation [27]. If adequate oral hygiene is maintained, bleeding on provocation will be minimal, and the enlarged gingiva will be firm and of a healthy pink color. Otherwise, the gingival area will be inflamed and the tissue will have a prominent red color.

The clinical characteristics of gingival hyperplasia in our patient were very similar to those described by Mangold and Salman [19,20]. A particularly pronounced gingival enlargement was noted in the area of the interdental papillae and attached gingiva. Concurring with Mangold, we also identified multiple erosions on the gingiva along with plaques [19]. Moreover, swab analyses confirmed the presence of the normal oral cavity microbiota, and the erosions subsided after topical application of Triamcinolone 0.1% in orabase.

Although whitish changes on the gingiva were initially suspected to be signs of hyperkeratosis, as similar lesions have been linked to vemurafenib therapy by other authors [28], pathohistological confirmation of hyperkeratosis was not obtained. The whitish lesions resembled those seen in white sponge nevus, a rare autosomal dominant condition characterized by extensive amounts of white, soft, and thick plaques on the oral mucosa [29]. Despite these similarities, the observed changes were confined solely to the gingiva and the patient had no family history of that condition. While the histopathological findings did not conclusively match the typical presentation of this condition, the clinical and histopathological features were most similar to a white sponge nevus-like lesion.

The ideal treatment for drug-induced gingival hyperplasia is discontinuation of the drug that has caused it and substituting it with a drug from another group, or reducing the dose of the drug that has given rise to this condition [30]. However, this is not always possible. Several malignant melanoma treatment protocols were prescribed to the patient in the focus of this report before vemurafenib introduction. None of those modalities were successful and no adequate alternative for vemurafenib was found. Moreover, as intentional therapy discontinuation or drug dose reduction could have endangered the patient’s life, none of these options was considered, as gingival hyperplasia was not deemed a valid reason to stop or modify therapy aimed at malignant melanoma management. However, as the patient was forced to discontinue vemurafenib use for one month due to shortages in local pharmacies, and he noticed marked improvements in gum condition, this may serve as anecdotal evidence supporting the view that complete vemurafenib cessation would resolve gingival hyperplasia. In practice, the vemurafenib dose was reduced and cobimetinib was added to the therapy protocol, which together with our treatment protocol, stabilized the gingiva condition.

Dental treatment of drug-induced gingival hyperplasia usually focuses on preventing or limiting excessive gingival tissue growth, where the patient’s oral hygiene status is the determining factor. Accordingly, non-surgical therapy is always advised in such cases, as its aim is suppressing the inflammatory component, thereby reducing the need for surgical treatment. As our patient also had periodontitis, the treatment protocol involved oral hygiene instructions and scaling and root planing.

In their research, Shephard et al. [31] reported significant improvement in gingival appearance following intensive periodontal treatment—scaling and root planing, extraction of teeth with considerable bone loss, and rigorous maintenance of oral hygiene—in a patient with metastatic melanoma undergoing vemurafenib therapy. The authors concluded that the marked improvement in gingival clinical characteristics following the periodontal intervention was likely due to the inflammatory processes associated with periodontal disease and their interaction with vemurafenib, which is an immune-modifying drug.

In a certain number of patients, including ours, after scaling and root planing, it is necessary to perform gingivectomy [32]. Although surgical removal of enlarged gingiva yields immediate and highly beneficial results, it cannot prevent reoccurrence of this condition, which is possible if the patient continues using the drug that gave rise to hyperplasia. Indeed, research shows that gingival hyperplasia can reemerge 3–6 months after surgical therapy, but remissions usually last at least 12 months [33]. Therefore, in some patients, periodic surgical reduction in enlarged gingiva is warranted and is the most optimal treatment course.

The treatment protocol in our study involved scaling and root planing, along with gingivectomy, which yielded favorable results. However, despite the initial improvements, during follow-up visits, persistent inflammation of the gingiva was observed and was attributed to suboptimal oral hygiene.

Two years into vemurafenib therapy, our patient started taking Cotellic (cobimetinib). This combined therapy aimed at inhibiting the signaling pathway of the V600E-mutated BRAF protein (vemurafenib), as well as the MEK enzyme, which is involved in the signaling pathway of both V600E- and V600K-mutated BRAF proteins (cobimetinib). The patient reported a complete resolution of gingival hyperplasia within two weeks of initiating Cotellic. Although a higher incidence of side effects compared to monotherapy is associated with this treatment protocol, Larkin et al. suggested that it reduces the incidence of keratoacanthomas and cutaneous squamous-cell carcinoma, alopecia, and arthralgias [34]. Long-term studies are nonetheless needed to evaluate the risks and toxicities associated with each agent when used in combination.

Gingival hyperplasia caused by vemurafenib is presently not a common occurrence. However, as this drug is increasingly being prescribed, the likelihood of encountering patients presenting with this condition in daily practice will also increase. Therefore, when treating patients with hyperplastic gingivitis, dentists must consider the side effects of vemurafenib as one of the possible causes.

## 4. Conclusions

Although BRAF and MEK inhibitors are effective in treating melanoma, the oral side effects associated with their use, such as gingival hyperplasia, are often overlooked when assessing their suitability for individual patients. While the nature and intensity of these adverse reactions can differ considerably, given their potential to undermine patients’ quality of life, leading to decreased adherence to treatment protocols, there is an evident need for a multidisciplinary strategy to prevent and mitigate potential complications.

## Figures and Tables

**Figure 1 jcm-14-00065-f001:**
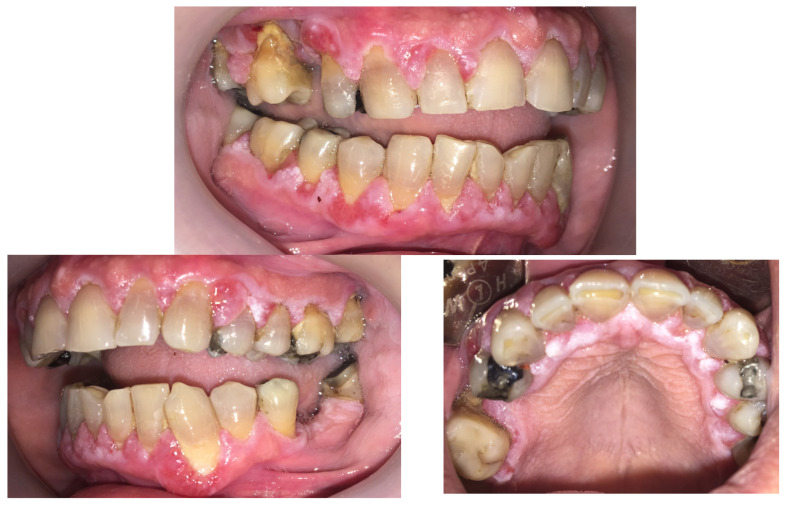
Gingival overgrowth during the first visit. The enlarged gums were covered with erosions and a thick, white plaque which appeared to be hyperkeratotic.

**Figure 2 jcm-14-00065-f002:**
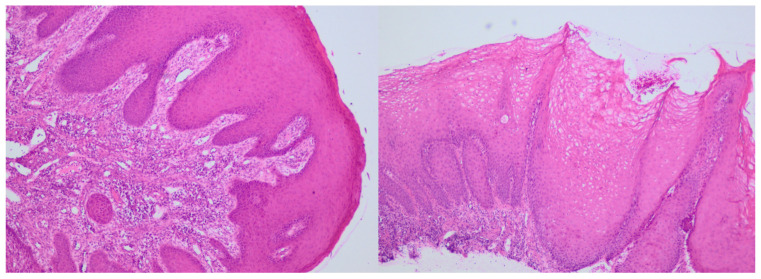
Histopathological changes in mucosa, pronounced acanthotic squamous cell epithelium with focal parakeratosis. Evidence of chronic inflammatory infiltrate in lamina propria (HEx40).

**Figure 3 jcm-14-00065-f003:**
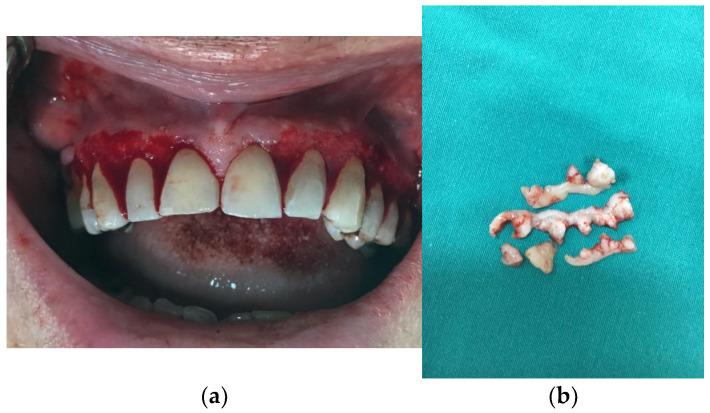
(**a**) Postoperative photograph of the patient’s mouth, (**b**) Eliminated hyperplastic gingiva.

**Figure 4 jcm-14-00065-f004:**
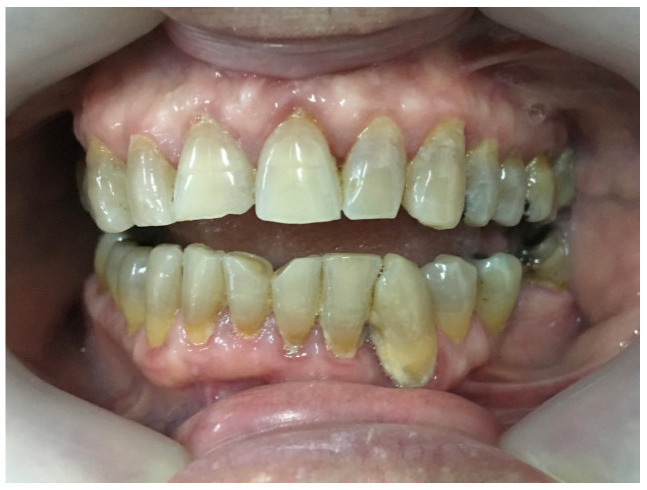
Tooth deposits and swollen inflamed gums three months after gingivectomy.

**Figure 5 jcm-14-00065-f005:**
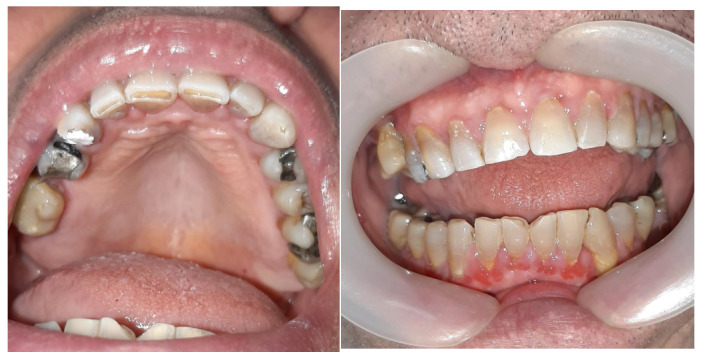
Photographs of the patient’s mouth after Cotellic was introduced. Inflamed but not hyperplastic gingiva was noted on examination.

## Data Availability

Data are contained within the article.

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
