# Peer review of "Impact of BRAF and MEK Inhibitors on Gingival Hyperplasia in Melanoma Patients—A Case Report"

_jcm, 2024, doi:10.3390/jcm14010065_

Round 1
Reviewer 1 Report
Comments and Suggestions for Authors
Impact of BRAF and MEK Inhibitors on Gingival Hyperplasia in
Melanoma Patients
Review Report
First of all, I would like to thank the editor for the opportunity to review this manuscript and the authors for submitting their paper.
In this manuscript, the authors, starting from a case report, present and discuss gingival hyperplasia as a possible adverse effect of melanoma systemic therapies (vemurafenib) and how it can impact the patient’s quality of life.
I offer the following suggestions to improve the manuscript:
Abstract:
Please provide a structured abstract to improve clarity and align with the journal's preferred
format and guidelines.
Case presentation:
• Line 87: You state, “Swabs for bacteria and fungi were negative.” Please specify
which microorganisms were tested and why.
• Line 98-99: “patient […] was scheduled for incisional biopsy.” How much time
elapsed between scaling root planning and the biopsy? This information would help
better assess the inflammatory infiltrate observed in the biopsy.
• Line 117: Please clarify whether Figure 4 corresponds to the 1-month control or the 3-
month control (when the “gingiva started swelling again”).
If available, it would be beneficial to include additional photos to create a more
comprehensive timeline documenting the evolution described in the text. Indicate the time
elapsed since surgical treatment in the captions. For example:
• 1 month (tooth deposits removed, edematous but no gingival overgrowth)
• 3 months (initial gingival overgrowth)
• 6 months (further gingival overgrowth)
• XY months (reduction in gingival enlargement observed after x time following the
modification of the therapy)
Author Response
Thank you for your detailed review and advice on the changes required to improve our manuscript further. We took into account most of the suggestions made and addressed them to the best of our knowledge. We hope that we succeeded in that and that you will find our paper worth publishing. Below, we respond in detail to each of the comments provided.
First of all, I would like to thank the editor for the opportunity to review this manuscript and the authors for submitting their paper.
In this manuscript, the authors, starting from a case report, present and discuss gingival hyperplasia as a possible adverse effect of melanoma systemic therapies (vemurafenib) and how it can impact the patient’s quality of life.
I offer the following suggestions to improve the manuscript:
- Abstract:
Please provide a structured abstract to improve clarity and align with the journal's preferred
format and guidelines.
We apologize for not formatting Abstract according to the journal guidelines. This issue has now been resolved and the Abstract content is now presented under the Background, Methods, Results, and Conclusions headings.
- Case presentation:
- Line 87: You state, “Swabs for bacteria and fungi were negative.” Please specify which microorganisms were tested and why.
Thank you for raising this important question. At our clinic, a swab is routinely taken for all patients that present with oral mucosal or gingival lesions. The same protocol was followed here and the subsequent microbial culture analyses confirmed presence of the normal oral cavity microbiota. (In the manuscript, the relevant sentence now states: “A swab was taken and the subsequent microbial culture analyses confirmed presence of the normal oral cavity microbiota.”)
- Line 98-99: “patient […] was scheduled for incisional biopsy.” How much time
elapsed between scaling root planning and the biopsy? This information would help better assess the inflammatory infiltrate observed in the biopsy.
The biopsy was taken one month after the scaling and root planing procedure. Based on the clinical characteristics of the gingiva at that moment, we concluded that enough time had passed for the exudative form of inflammation to subside, while local therapy had no impact on the whitish changes on the gingiva which we suspected to be hyperkeratosis.
- Line 117: Please clarify whether Figure 4 corresponds to the 1-month control or the 3-
month control (when the “gingiva started swelling again”).
Figure 4 provides oral cavity images taken three months after gingivectomy.
If available, it would be beneficial to include additional photos to create a more
comprehensive timeline documenting the evolution described in the text. Indicate the time
elapsed since surgical treatment in the captions. For example:
- 1 month (tooth deposits removed, edematous but no gingival overgrowth)
- 3 months (initial gingival overgrowth)
- 6 months (further gingival overgrowth)
- XY months (reduction in gingival enlargement observed after x time following the
modification of the therapy)
We fully agree that the order of the displayed images the reviewer proposed would give the readers a better insight into this case presentation. Unfortunately, due to the underlying disease, the patient was unable to attend regular follow-up examinations. Hence, we had to make the most suitable selection among the photos that were available to us.
Reviewer 2 Report
Comments and Suggestions for Authors
- The authors affirm that the swabs for bacteria and fungi were negative. The oral environment always contains at least bacteria. Explain how this result has been obtained.
- Describe on what side the biopsy has been taken.
- Informe consent signed by the patients to use his data is mandatory.
- Explain the reasons for performing a gingivectomy when the cases published before describing that the reducing of the doses produce an improvement of the gingival enlargement and is this case this data was described.
- Previous paper described that a marked improvement in gingival clinical characteristics following the periodontal intervention has been obtained. The authors treat this condition with oral hygiene instructions and remove tooth deposits. Did the patients suffer from periodontitis? If this was the case, at least scaling and root planning is the appropriate treatment.
Author Response
We wish to express our sincere gratitude to the reviewer for providing us with insightful and constructive feedback, which was instrumental for elevating the quality of our manuscript. Below, we respond in detail to each of the comments provided.
1-The authors affirm that the swabs for bacteria and fungi were negative. The oral environment always contains at least bacteria. Explain how this result has been obtained.
Thank you for raising this important question, as it indicates that our original statement was unclear. In practice, a swab was taken, as is a common practice at our clinic when patients present with oral mucosal or gingival lesions. The culture analyses confirmed presence of the normal oral cavity microbiota. (In the manuscript, the relevant sentence now states: “A swab was taken and the subsequent microbial culture analyses confirmed presence of the normal oral cavity microbiota.”)
2- Describe on what side the biopsy has been taken.
We appreciate this request. In the revised manuscript, we now state: “Hyperplastic gingiva samples taken from two sites—the interdental papilla from the vestibular side in the area of ​​tooth 16 and from the palatal side of the upper front teeth—was subjected to pathohistological analysis.”
3- Informe consent signed by the patients to use his data is mandatory.
We concur with this observation and apologize for not explicitly stating that these measures were taken as a part of this research. Indeed, prior to commencing the study, the patient provided a written informed consent after receiving detailed verbal and written explanation regarding the nature of the study and the treatment he would undergo. (all relevant documentation has been submitted to the journal and was found acceptable)
4- Explain the reasons for performing a gingivectomy when the cases published before describing that the reducing of the doses produce an improvement of the gingival enlargement and is this case this data was described.
We appreciate these valuable observations and concur with the reviewer. In response, we have now added the following text to our manuscript:
“Drug-induced gingival hyperplasia can be eliminated by discontinuing the drug that has caused it and substituting it with a drug from another group or by reducing the dose of the drug that has given rise to this condition. However, this is not always possible. Several malignant melanoma treatment protocols were prescribed to the patient in focus of this report before vemurafenib introduction. None of those modalities were successful and no adequate alternative for vemurafenib was found. Moreover, as intentional therapy discontinuation or drug dose reduction could have endangered the patient’s life, none of these options was considered, as gingival hyperplasia was not deemed a valid reason to stop or modify therapy aimed at malignant melanoma management. However, as the patient was forced to discontinue vemurafenib use for one month due to shortages in local pharmacies, and he noticed marked improvements in gum condition, this may serve as anecdotal evidence supporting the view that complete vemurafenib cessation would resolve gingival hyperplasia. In practice, vemurafenib dose was reduced and cobimetinib was added to the therapy protocol, which together with our treatment protocol, stabilized the gingiva condition.”
5- Previous paper described that a marked improvement in gingival clinical characteristics following the periodontal intervention has been obtained. The authors treat this condition with oral hygiene instructions and remove tooth deposits. Did the patients suffer from periodontitis? If this was the case, at least scaling and root planning is the appropriate treatment.
We appreciate this question and have now expanded our case description in the Case presentation section as well as the Discussion section where we indicate that, as our patient also had periodontitis, the treatment protocol involved oral hygiene instructions and scaling and root planning.
Reviewer 3 Report
Comments and Suggestions for Authors
This study aims to analyze the development of gingival hyperplasia in patients with melanoma undergoing therapy with genetic inhibitors. The study is interesting and quite well described, it is a case report that certainly addresses a topic not much analyzed in the literature.
I advise the authors to provide some more details in the materials and methods section regarding the therapy used and the protocols followed by the patient, for the resolution of the gingival problem. Improve the discussion with the following article: "Manuelli M., Marcolina M., Nardi N., Bertossi D., de Santis D., Ricciardi G., Luciano U., Nocini R., Mainardi A., Lissoni A., Abati S., et al. Oral mucosal complications in orthodontic treatment (2019) Minerva Stomatologica, 68 (2), pp. 84 - 88. DOI:10.23736/S00264970.18.04127-4", which highlights the oral flora of the oral cavity is also influenced by other situations, not only tumoral and that often it is however the multidisciplinary approach to improve such situations.
I recommend rechecking the English language.
Author Response
We would like to take this opportunity to thank the reviewer for such a positive assessment of our work and for providing us with valuable suggestions for improving our manuscript further.
This study aims to analyze the development of gingival hyperplasia in patients with melanoma undergoing therapy with genetic inhibitors. The study is interesting and quite well described, it is a case report that certainly addresses a topic not much analyzed in the literature.
- I advise the authors to provide some more details in the materials and methods section regarding the therapy used and the protocols followed by the patient, for the resolution of the gingival problem.
Thank you for your valuable suggestion. In response, in the Case presentation section we have added more detailed information on the therapy and protocols used during our study. We hope that in so doing we have adequately addressed your concerns. We specifically state:
“Due to the presence of both periodontal disease and gingival hyperplasia with abundant supragingival and subgingival dental plaque, we commenced treatment with scaling and root planing. As indicated in the revised manuscript, the procedure was performed under local anesthesia (2% lidocaine with adrenaline 1:100.000) using periodontal curettes (Gracey Access curettes, Kohler, Austria) and ultrasonic scalers (Mini Piezon, Electro-Medical Systems, Nyon, Switzerland). The therapy was carried out on two consecutive days within 24 hours, starting with the right maxillary and mandibular quadrants, without the use of antibiotics or antiseptics.
At the next follow-up examination four weeks later, it was evident that the inflammation had decreased. However, as gingiva remained hyperplastic, the decision was made to surgically remove the enlarged tissue. Gingivectomy was performed under local anesthesia, during which the hyperplastic gingiva was removed, starting with the vestibular and palatal side of the upper jaw, followed by the lower jaw. A COE-PAK™ surgical dressing was placed over all surgical sites, and was removed at the control visit seven days later. The intervention was performed without any complications and the postoperative course was uneventful.”
Improve the discussion with the following article: "Manuelli M., Marcolina M., Nardi N., Bertossi D., de Santis D., Ricciardi G., Luciano U., Nocini R., Mainardi A., Lissoni A., Abati S., et al. Oral mucosal complications in orthodontic treatment (2019) Minerva Stomatologica, 68 (2), pp. 84 - 88. DOI:10.23736/S00264970.18.04127-4", which highlights the oral flora of the oral cavity is also influenced by other situations, not only tumoral and that often it is however the multidisciplinary approach to improve such situations.
We greatly appreciate the reviewer’s suggestion to incorporate this literature source into our article. It is referenced in the Discussion section when describing the factors that contribute to the gingival hyperplasia development.
I recommend rechecking the English language.
We sincerely apologize for any issues with the English language in the original version of our manuscript. We have since revised the content thoroughly, and hope that no errors in grammar, style or terminology use remain.
Reviewer 4 Report
Comments and Suggestions for Authors
I have read the article with great interest. The manuscript addresses a rare side effect but important for patient’s quality of life.
It is well organized, written in a clear, accessible style including not only the diagnosis but also the management of the patient through the years.
I believe the manuscript would benefit from more information about the melanoma staging at the first diagnosis, wide local excision/SLNB etc and the morphology-distribution of the rash that the patient developed after the treatment etc, if these data are available.
Author Response
First, we would like to thank the reviewer for the time and effort dedicated to the evaluation of our manuscript. It is encouraging that you have found our study relevant and that your assessment of the way we presented our case was positive. We also appreciate all the critique we received and have addressed each comment to the best of our ability, aiming to enhance the quality of our paper further.
I have read the article with great interest. The manuscript addresses a rare side effect but important for patient’s quality of life.
It is well organized, written in a clear, accessible style including not only the diagnosis but also the management of the patient through the years.
I believe the manuscript would benefit from more information about the melanoma staging at the first diagnosis, wide local excision/SLNB etc and the morphology-distribution of the rash that the patient developed after the treatment etc, if these data are available.
Thank you for requesting this information, as it will certainly be of value for the journal readers. As our patient’s medical records contained the following information, it was included in the revised manuscript:
- At the time of the first diagnosis, melanoma staging was Breslow thickness of 4 mm and Clark level IV.
- The dermatologist described the rash as erythema on the face, neck, nape, and trunk, with both single and diffuse macules and papules, concluding that the clinical presentation is consistent with a toxoallergic exanthem.
Reviewer 5 Report
Comments and Suggestions for Authors
General Comment:
The manuscript provides a well-structured case report highlighting the oral side effects of BRAF and MEK inhibitors in a melanoma patient. The clinical details are comprehensive, and the discussion appropriately contextualizes the findings within the existing literature. However, minor revisions are needed to enhance clarity and consistency in the presentation.
Recommendation: Minor Revision
The following revisions are suggested:
1. Title (Line 2-3): Clearly mention the study type (e.g., “A Case Report”) in the title.
2. Abstract (Line 13): Avoid using parentheses within the abstract, such as “(e.g., vemurafenib).”
3. Keywords (Line 22): Expand the keyword list to include terms like “case report”.
4. Case Report Section: Include comparisons to baseline gingival tissue or known patterns of hyperplasia induced by similar therapies.
Author Response
We want to take this opportunity to thank the reviewer for such a positive assessment of our work and for providing us with advice on how to further improve our manuscript. We have addressed all the issues raised and hope that these changes will be adequate, rendering this revised version suitable for publication.
The manuscript provides a well-structured case report highlighting the oral side effects of BRAF and MEK inhibitors in a melanoma patient. The clinical details are comprehensive, and the discussion appropriately contextualizes the findings within the existing literature. However, minor revisions are needed to enhance clarity and consistency in the presentation.
Recommendation: Minor Revision
The following revisions are suggested:
- Title (Line 2-3): Clearly mention the study type (e.g., “A Case Report”) in the title.
We appreciate this valuable suggestion and have now revised the article title to include “A Case Report.”
- Abstract (Line 13): Avoid using parentheses within the abstract, such as “(e.g., vemurafenib).”
We appreciate this valuable advice and have removed all parentheses from the Abstract content.
- Keywords (Line 22): Expand the keyword list to include terms like “case report”.
We wish to thank the reviewer for this valuable suggestion and have included “case report” into the keyword list.
- Case Report Section: Include comparisons to baseline gingival tissue or known patterns of hyperplasia induced by similar therapies.
We are grateful for this valuable advice and have addressed it by expanding the Discussion section to compare our findings with those published in pertinent literature.
Round 2
Reviewer 2 Report
Comments and Suggestions for Authors
The requeriments have been attended
Author Response
Thank you once again for showing interest in our manuscript, as well as for helping us to improve it further.